# Democratizing LLMs for Low-Resource Languages by Leveraging their English Dominant Abilities with Linguistically-Diverse Prompts

## ABSTRACT

Large language models (LLMs) are known to effectively perform tasks by simply observing few exemplars. However, in low-resource languages, obtaining such hand-picked exemplars can still be challenging, where unsupervised techniques may be necessary. Moreover, competent generative capabilities of LLMs are observed only in high-resource languages, while their performances among under-represented languages fall behind due to pre-training data imbalance. To elicit LLMs' ability onto low-resource languages without any supervised data, we propose to assemble synthetic exemplars from a diverse set of high-resource languages to prompt the LLMs to translate from any language into English. These prompts are then used to create intra-lingual exemplars to perform tasks in the target languages. Our unsupervised prompting method performs on par with supervised few-shot learning in LLMs of different sizes for translations between English and 13 Indic and 21 African low-resource languages. We also show that fine-tuning a 7B model on data generated from our method helps it perform competitively with a 175B model. In non-English translation tasks, our method even outperforms supervised prompting by up to 3 chrF++ in many low-resource languages. When evaluated on zero-shot multilingual summarization, our method surpasses other English-pivoting baselines by up to 4 ROUGE-L and is also favored by GPT-4.

## 1 INTRODUCTION

Recent scaling effort in foundation large language models (Brown et al., 2020; Chowdhery et al., 2022; Scao et al., 2022; Touvron et al., 2023) with massive pre-training data has enabled them to learn a broad range of natural language tasks through few-shot in-context learning, where a few input-output exemplars are concatenated to the test input to prompt the model to predict the output and no gradient update of the model is performed. While most LLMs are pre-trained with multilingual corpora in addition to the gigantic English corpus, and have been shown to demonstrate impressive abilities in other languages (Brown et al., 2020; Chowdhery et al., 2022; Scao et al., 2022), they only excel in high-resource languages, such as French. Further, they may still require pivoting the inputs into English, that is, performing tasks in English first before reverting the response back to native language outputs (Shi et al., 2022; Huang et al., 2023). Improving LLMs' abilities in extremely low-resource languages can be even more challenging, particularly where the data coverage is less than 0.0001% (Scao et al., 2022) or none at all (Touvron et al., 2023). We also found that the models may confusedly generate a wrong language and struggle to process low-resource non-latin scripts due to overly fragmented tokenization, where short texts are broken into extremely long byte-level tokens.

In this work, we focus on unsupervised translation and summarization tasks in low-resource languages, where no few-shot exemplars are available. We primarily focus only on foundation LLMs (Scao et al., 2022; Touvron et al., 2023) to maximally avoid leakage of human-annotated data inherent in instruction-tuned models (Ouyang et al., 2022; Muennighoff et al., 2022). To this end, we propose Linguistically-Diverse Prompting (LDP), a technique that promotes an LLM to locate the task of "translating any language $X$ into English" by showing the model exemplar pairs between *every* language and English (En). Practically, we gather a small set of synthetic $X{\rightarrow}$En exemplars from a diverse set of high-resource languages using an off-the-shelf unsupervised MT model (Tran et al., 2020). To ensure disversity, languages with script types ranging from Latin (Fr) to Chinese (Zh)

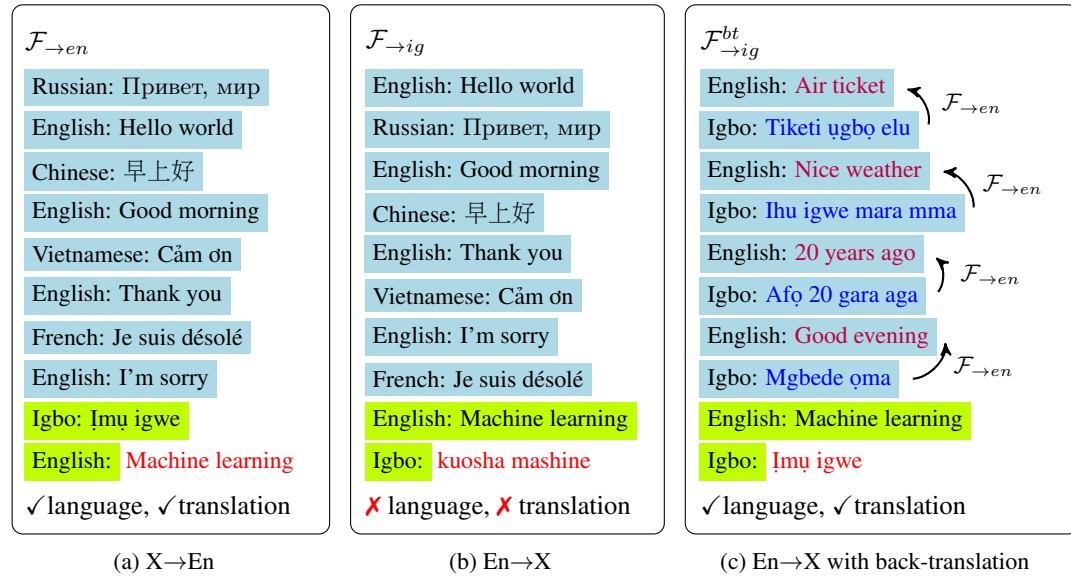

Figure 1: LDP prompting examples: (1a) $\mathcal{F}_{\rightarrow en}$ translates from any language into English by concatenating the fixed linguistically-diverse in-context exemplars and input text to prompt LLMs to generate the correct translation. (1b) Similar $\mathcal{F}_{\rightarrow ig}$ translates English into Igbo, but with low accuracy. (1c) $\mathcal{F}_{\rightarrow ig}^{bt}$ translates English to Igbo using synthetic intra-lingual exemplars generated from unlabeled monolingual target-language data with $\mathcal{F}_{\rightarrow en}$.

characters are used. An example of the method is shown in Figure 1. Our method is based on the following empirical observations of LLMs: (*i*) in-context exemplars may play a larger role in helping the model *locate* the task in its pre-trained knowledge (Xie et al., 2021), (*ii*) LLMs possess dominant abilities in English while lagging behind in other lower-resource languages (Ouyang et al., 2022; Touvron et al., 2023; Huang et al., 2023; Shi et al., 2022).

Our method is shown to translate any low-resource language into English with quality on par with supervised prompting, which allows us to build intra-lingual exemplars with unlabeled data to prompt the models to translate into low-resource languages. In our experiments with BLOOM (Scao et al., 2022) and InstructGPT (text-davinci-003) (Ouyang et al., 2022), our unsupervised LDP method performs on par with supervised prompting in $X\rightarrow$En and En$\rightarrow X$ translation tasks across 13 Indic and 21 African low-resource languages. Furthermore, adapting our method to $X\rightarrow Y$ non-English directions even outperforms supervised promptings by up to 3 chrF++ in pairs involving low-resource languages. In multilingual summarization tasks (Narayan et al., 2018), our zero-shot LDP method outperforms both basic prompting and other English-pivoting methods by up to 4 ROUGE-L and is generally favored by GPT-4-EVAL (Liu et al., 2023), demonstrating good human alignment.

## 2 RELATED WORK

Large language models (LLMs) display outstanding capabilities because they are pre-trained on massive amounts of internet text data (Radford et al., 2019; Brown et al., 2020; Chowdhery et al., 2022; Scao et al., 2022; Touvron et al., 2023). Without any gradient update, foundation LLMs are able to perform in-context few-shot learning by simply observing a list of high-quality input-output exemplars (Brown et al., 2020; Wei et al., 2023). This technique works across a broad range of tasks that involve language understanding, reasoning and generation (Brown et al., 2020; Wei et al., 2022; Shi et al., 2022). Much research has been conducted to understand in-context learning. Some suggest that the models secretly perform gradient descent on the exemplars (Dai et al., 2022), while others demonstrate that most of the knowledge is learned during pre-training, and that in-context exemplars are only to provide evidence for the model to *locate* the intended task via a Bayesian inference process (Xie et al., 2021; Min et al., 2022; Zhou et al., 2023).

Most LLMs are trained with multilingual corpora (Wenzek et al., 2020), albeit just a tiny fraction of the large English corpora (Radford et al., 2019; Brown et al., 2020). Despite this imbalance, LLMs still exhibit strong multilingual capabilities in high-resource languages like French, German and Chinese, often with the help of supervised translation systems (Shi et al., 2022) or prompting the model to firstly generate intermediate English text before arriving at the final answer (Huang et al., 2023). BLOOM (Scao et al., 2022) is one of the LLMs trained with the most number of languages, whose ROOTS corpus consists of data from 46 languages (Laurençon et al., 2022). The ROOTS corpus includes 34 Indic and African languages regarded as low-resource, with each language having a pre-training coverage of less than 1% in Hindi for the Indic group, to $2e^{-5}$% in Tumbuka for the African group, as shown in Figure 7 in the Appendix. We use BLOOM as the main model to evaluate our methods and baselines in the 34 low-resource languages. Our linguistically-diverse prompting strategy is also an English-pivoting method, but it is different from other cross-lingual counterparts (Shi et al., 2022; Huang et al., 2023) in that while others only pivot inputs to English intermediates, we use in-context pairs between English and a diverse set of high-resource languages to promote the intended task in the target low-resource language.

Our work also intersects with unsupervised multilingual machine translation (UMT), where back-translation (Edunov et al., 2018; Lample et al., 2018; Conneau & Lample, 2019; Liu et al., 2020; Nguyen et al., 2022b), along with other methods like bi-text mining (Tran et al., 2020; Nguyen et al., 2022a), are proven to be effective. English-pivoting is also prominent in the realm of machine translation, where training models on high-resource En$\leftrightarrow X$ bitext improves lower-resource En$\leftrightarrow Y$ tasks (Garcia et al., 2020; 2021). Nonetheless, the noteworthy gap between existing UMT and LLM models is that their language coverages do not overlap much, preventing us from using UMT to enhance LLMs. For example, BLOOM does not cover 14 languages supported by CRISS (Tran et al., 2020).

Analyses of machine translation using LLMs have also been done. Hendy et al. (2023) show that GPT models can perform competitively alongside the best MT models. Zhu et al. (2023) focus on optimizing supervised exemplars selection strategies. Sia & Duh (2023) discover that using specific coherent prompts for each input helps improve performance. Nonetheless, such studies only consider supervised instruction-tuned models (Ouyang et al., 2022; Muennighoff et al., 2022), which may risk test-set contamination (Muennighoff et al., 2022). Thus, there is still limited research involving low-resource languages in completely zero-shot setups. As such, since low-resource languages may not enjoy the privilege of having large unlabeled data to conduct searching, only random selection is used in this study, while optimal exemplar selection is not within the scope of the paper.

## 3 METHOD

### 3.1 LINGUISTICALLY-DIVERSE PROMPTING (LDP)

Our method is inspired from two empirical observations: (*i*) The first is that LLMs have already learned most of the task concepts implicitly during pre-training, and that in-context exemplars play a larger role in providing evidence for the model to identify the intended task (Xie et al., 2021; Min et al., 2022; Zhou et al., 2023). (*ii*) The second observation is that LLMs perform generative tasks particularly well in only a handful of major languages (English and other high-resource languages), where pre-training data is significantly larger than other languages (Brown et al., 2020; Touvron et al., 2023; OpenAI, 2023). To achieve better performance on lower-resource languages, it has been shown that we may need to instruct the LLMs to generate intermediate reasoning in English before producing the final answers in the target language (Huang et al., 2023), or alternatively to translate non-English inputs to English and perform tasks in English entirely (Shi et al., 2022).

Figure 1 illustrates how our LDP method aims to take advantage of the aforementioned observations. Particularly, we prompt the model to identify the task of "translating from *any language $X$* into $E$", by demonstrating pairs from "every language" to $E$. In practice, as shown in Figure 1a, we use synthetic pairs from diverse high-resource languages as in-context exemplars to prompt the models to translate the target low-resource language $X$ (*e.g.,* Igbo) into English (En) with high quality. Such a diverse set of prompt languages would include various script types ranging from Latin alphabets to logograms. Figure 1b shows that applying the same technique for En$\rightarrow X$ task may result in incorrect translations. In Figure 1c, we leverage LDP to translate unlabeled texts of target $X$ language into En, forming back-translated synthetic pairs to prompt the model to translate from En to $X$ with higher

(a) LDP for translation for $X{\rightarrow}$En, En${\rightarrow}X$ and $X{\rightarrow}Y$.

(b) LDP for summarization.

Figure 2: Illustrations of adopting LDP for $X{\rightarrow}$En, En${\rightarrow}X$ and $X{\rightarrow}Y$ translation (2a) and summarization (2b). For $X{\rightarrow}$En, the colored box [z] represents an unlabeled text in language z, [en] represents its corresponding En translation, while [x] stands for the test input in language x and uncolored box [ên] represents model outputs. For En${\rightarrow}X$, $[\text{en}^{\text{x}}]$ represents En text translated with $\mathcal{F}^{mt}_{\text{x}\rightarrow\text{en}}$. For $X{\rightarrow}Y$, $[\overline{\text{y}}^{\text{en}}]$ represents a text in language y translated from En text $[\hat{\text{en}}^{\text{x}}]$. Similarly for summarization (2b), $[\hat{\text{s}}_{\text{z}}]$ represents a summary in language z for document $[\text{d}_{\text{z}}]$.

quality. This is because the target-side prompt distribution is now realistic and consistently close to the true target distribution, which has been shown to be crucial for in-context learning (Xie et al., 2021).

### 3.2 LDP FOR TRANSLATION TASKS

We adopt LDP in translation tasks for $X \rightarrow E$, $E \rightarrow X$ and $X \rightarrow Y$ (where $X, Y \neq E$), differently, as demonstrated in Figure 2a, where we assume $E =$ English (En) for better comprehensibility.

$X \rightarrow E$ **task.** As mentioned above, we first gather $n$ $Z_i{\rightarrow}E$ exemplar pairs $(s_{Z_i}, t^i_E)$ with $Z_i \in \mathcal{Z}$ and $\mathcal{Z}$ being a diverse set of languages with various writing systems and dissimilar lexical and regional characteristics, such as Chinese (Zh), and $Z_i \notin \{X, E\}$. Such exemplars can be collected from unlabeled data of the respective language $Z_i$ and using unsupervised MT models to translate them into $E$. From that, we can perform translation of an input $s_X$ into $E$ with an LLM ($\theta$) by conditioning the LDP prompts as:

$$\mathcal{F}^{mt}_{X\rightarrow E}(s_X) \sim p_\theta(y|s_X, s_{Z_1}, t^1_E, .., s_{Z_n}, t^n_E) \quad (1)$$

$E \rightarrow X$ **task.** We leverage $\mathcal{F}^{mt}_{X\rightarrow E}$ to build intra-lingual prompts with unlabeled data from the target language $X$. Specifically, given $m$ unlabeled texts $s^j_X \in \mathcal{D}_X$ with $\mathcal{D}_X$ being a monolingual corpus in language $X$, we produce synthetic back-translation (BT) target $s^j_E = \mathcal{F}^{mt}_{X\rightarrow E}(s^j_X)$. Then, we use the BT synthetic pairs as exemplars for $E \rightarrow X$ tasks for a test input $s_E$:

$$\mathcal{F}^{mtbt}_{E\rightarrow X}(s_E) \sim p_\theta(y|s_E, s^1_E, t^1_X, ..., s^m_E, t^m_X) \quad (2)$$

The intra-lingual exemplars with the same language in the target side helps the model locate the intended language to generate more effectively than a standard language tag, as these exemplars show the model what the intended language looks like.

Note that we could also use $\mathcal{F}^{mtbt}$ for $X \rightarrow E$ ($\mathcal{F}^{mtbt}_{X\rightarrow E}$) by simply swapping the direction of the $(s^j_E, t^j_X)$ to $(s^j_X, t^j_E)$. However, we found in the experiments that both $\mathcal{F}^{mt}$ and $\mathcal{F}^{mtbt}$ perform similarly and on par with supervised prompting for the $X \rightarrow E$ task, suggesting that we do not need any supervised or unlabeled data to translate any language into English. Furthermore, in Section 4.4, we demonstrate that we can even omit these back-translation exemplars entirely with non-BT $\mathcal{F}^{mt}$ LDP by using native language tags.

$X \rightarrow Y$ **task.** We leverage $\mathcal{F}^{mtbt}_{X\rightarrow E}$ and $\mathcal{F}^{mtbt}_{E\rightarrow X}$ to build $E$-pivoting triplets from unlabeled text from the source side. Specifically, given unlabeled text $s^j_X \in \mathcal{D}_X$ in $X$ language, we back-translate them into $s^j_E = \mathcal{F}^{mtbt}_{X\rightarrow E}(s^j_X)$, which we then use to produce $s^j_Y = \mathcal{F}^{mtbt}_{E\rightarrow Y}(s^j_E)$. This process forms

Table 1: Averaged performances of different prompting techniques across various model sizes and types, namely BLOOM (Scao et al., 2022) and InstructGPT text-davinci-003 (Brown et al., 2020; Ouyang et al., 2022), in translation tasks between English (En) and 13 Indic (Indic13) and 21 African (Afri21) low-resource languages present in the ROOTS corpus (Laurençon et al., 2022).

| | Indic13-En | | En-Indic13 | | Afri21-En | | En-Afri21 | |
|---|---|---|---|---|---|---|---|---|
| | chrF++ | BLEU | chrF++ | BLEU | chrF++ | BLEU | chrF++ | BLEU |
| **Foundation BLOOM-175B** | | | | | | | | |
| Supervised-8-shot | 47.31 | 22.32 | 34.66 | 9.02 | 28.64 | 8.35 | 14.93 | 2.00 |
| Unsupervised-LDP | 47.62 | 22.38 | 34.54 | 8.88 | 28.72 | 8.71 | 14.57 | 1.89 |
| **Foundation BLOOM-7B** | | | | | | | | |
| Supervised-8-shot | 39.86 | 14.77 | 24.02 | 4.42 | 21.51 | 4.33 | 11.27 | 0.59 |
| Unsupervised-LDP | 39.88 | 14.96 | 24.41 | 4.52 | 20.47 | 3.65 | 12.04 | 0.62 |
| Fine-tune QKV (2B params) | 42.19 | 17.13 | 32.72 | 8.33 | 21.14 | 5.15 | 15.73 | 2.13 |
| **Supervised RLHF InstructGPT (text-davinci-003)** | | | | | | | | |
| Zero-shot with instruction | 35.37 | 11.48 | 20.71 | 3.88 | 27.10 | 8.04 | 15.45 | 1.13 |
| Supervised-6-shot | 37.07 | 13.13 | 24.74 | 5.21 | 31.51 | 10.88 | 19.22 | 2.66 |
| Unsupervised-LDP | 38.45 | 14.22 | 25.17 | 5.06 | 31.92 | 11.12 | 19.51 | 2.61 |
| **Supervised upperbound** | | | | | | | | |
| NLLB-200 distilled | *61.00* | *37.24* | *46.77* | *18.78* | *48.42* | *26.92* | *39.18* | *12.95* |

triplets $[s_X^j, s_E^j, s_Y^j]$ to prompt the model to generate intermediate $E$ translation before producing the final result in $Y$. Formally, given an input $s_X$, the translation in $Y$ is computed as:

$$\mathcal{F}_{X \to Y}^{mt}(s_X) \sim p_\theta(y|s_X, s_X^1, s_E^1, s_Y^1, ..., s_X^m, s_E^m, s_Y^m) \qquad (3)$$

**Unsupervised fine-tuning.** The $\mathcal{F}_{X \to E}^{mt}$ prompting method also allows us to create larger-scale synthetic $X$-$E$ data from unlabeled corpora to fine-tune the model for translation tasks without any in-context prompt at inference time. Specifically, we use the `[input]<lang-tag>[output]` template to construct multilingual training samples with the generated data pairs from multiple low-resource languages. We fine-tune the query-key-value linear weights of all attention layers, which account for 20-30% of the total parameters to avoid overfitting.

## 3.3 LDP FOR MULTILINGUAL SUMMARIZATION

For summarization tasks, we extend XLT (Huang et al., 2023), a recent English-pivoting cross-lingual prompting technique, with document-summarization pairs from diverse high-resource non-English languages. Our technique is illustrated in Figure 2b. Formally, given an input document $d_X$ in the target language $X$ and $n$ unlabeled documents $d_{Z_i}$ with $Z_i \in \mathcal{Z}$ and $\mathcal{Z}$ being a diverse set of high-resource non-English languages, we use zero-shot XLT with English-pivoting instructions to generate summarization $s_{Z_i} = \text{XLT}(d_{Z_i})$. We then use the synthetic document-summary pairs $(d_{Z_i}, s_{Z_i})$ as LDP in-context exemplars to compute the target-language summary for $d_X$ as:

$$\mathcal{F}_X^{sum}(d_X) \sim p_\theta(y|d_X, d_{Z_1}, s_{Z_1}, .., d_{Z_n}, s_{Z_n}) \qquad (4)$$

Similar to $E \to X$ translation task, we then use zero-shot $\mathcal{F}_X^{sum}$ to generate synthetic intra-lingual prompts from $m$ unlabeled documents $d_X^j \in \mathcal{D}_X$ by producing summary $s_X^j = \mathcal{F}_X^{sum}(d_X^j)$ in $X$ language. After that, we compute the final summary for the input $d_X$ with $\hat{\mathcal{F}}_X^{sum}$ as:

$$\hat{\mathcal{F}}_X^{sum}(d_X) \sim p_\theta(y|d_X, d_X^1, s_X^1, .., d_X^m, s_X^m) \qquad (5)$$

## 4 EXPERIMENTS

In this section, we evaluate our method in various translation tasks (Sections 4.1 and 4.2) and summarization tasks (4.3) across different settings and languages. We also conduct extensive analyses to provide further insights into our method (4.4).

### 4.1 LOW-RESOURCE ↔ ENGLISH TRANSLATION

As the ROOTS corpus (Laurençon et al., 2022) on which BLOOM (Scao et al., 2022) was pre-trained offers the most diverse language coverage with open-sourced transparency, we tested our methods

Table 2: chrF++ translation scores for X to Y non-english directions across high-high, high-low and low-low languages groups.

| | High-High | | High-Low | | | | Low-Low | | | |
|---|---|---|---|---|---|---|---|---|---|---|
| | Vi-Fr | Fr-Vi | Zh-Ne | Ne-Zh | Es-Pa | Pa-Es | Ta-Sw | Sw-Ta | Te-Sw | Sw-Te |
| **Foundation BLOOM-175B** | | | | | | | | | | |
| Supervised-8-shot | 52.17 | 51.50 | 30.91 | 17.83 | 25.67 | 37.71 | 31.45 | 31.81 | 31.46 | 25.84 |
| Unsupervised-LDP | 52.66 | 50.24 | 31.61 | 18.34 | 27.85 | 39.51 | 34.61 | 34.47 | 32.14 | 30.57 |
| **Supervised InstructGPT (text-davinci-003)** | | | | | | | | | | |
| XLT (Huang et al., 2023) | 51.16 | 44.84 | 28.56 | 13.26 | 23.61 | 34.18 | 24.20 | 25.46 | 24.89 | 23.48 |
| Unsupervised-LDP | 51.19 | 45.80 | 28.67 | 15.80 | 25.40 | 35.02 | 27.24 | 27.70 | 28.95 | 25.12 |

mainly with the BLOOM model on 13 Indic (Indic13) languages and 21 African (Afri21) languages present in the ROOTS corpus. We also conduct experiments with supervised InstructGPT (text-davinci-003) (Ouyang et al., 2022) to provide further references. As it is not publicly disclosed how large text-davinci-003 is or whether it was trained on the test sets, its results are only for a comparison of prompting techniques within the InstructGPT section. For each of the 68 language pairs, we sample randomly and evaluate 200 sentences from each test set with the same seed to limit the cost of API calls[1]. Following Costa-jussà et al. (2022), we report results in mainly chrF++ score (Popović, 2015), which is a universal metric for all languages, while also reporting SacreBLEU (Post, 2018) as a complementary metric. The full list of languages and other details are provided in the Appendix.

In terms of methodologies, for supervised prompting, we collect as many supervised pairs as the models can fit within their context lengths (8 for BLOOM and 6 for GPT davinci-003). We use `<src>[input]\n<tgt>[output]` as the prompt template, where `<src>` and `<tgt>` are the language tag names in English. For our unsupervised linguistically-diverse prompting (LDP) method, we use 4 LDP $Z_i \leftrightarrow$En pairs from Arabic (Ar), Chinese (Zh), Vietnamese (Vi) and French (Fr) to conduct $X \rightarrow E$ synthetic data generation with $\mathcal{F}^{mt}_{X \rightarrow E}$ before using them as intra-lingual prompts for the target pair with $\mathcal{F}^{mtbt}_{X \leftrightarrow E}$, as explained in Section 3. For LDP, we do not include the language tags in the prompts as they offer no benefit. In our fine-tuning experiment, we use $\mathcal{F}^{mt}_{X \rightarrow E}$ to generate synthetic training data from various unlabeled sources (Wenzek et al., 2020) to fine-tune BLOOM-7B.

Table 1 shows the averaged chrF++ and BLEU scores for translations between English and 13 Indic and 21 African low-resource languages across different prompting techniques with BLOOM-175B, BLOOM-7B and GPT text-davinci-003 models. The first noticeable finding is that our unsupervised-LDP method performs on par with supervised prompting across all language groups and LLM models. This indicates that the synthetic prompts generated by our $\mathcal{F}^{mt}_{X \rightarrow E}$ technique are as good as supervised prompts for this purpose,[2] thanks to the LLMs' outstanding generative ability in English. Furthermore, fine-tuning a 7B model with data generated by itself helps the model to advance towards the performance of its 175B counterpart, especially for En$\rightarrow X$ direction. This suggests that fine-tuning the model on more low-resource language data improves generative abilities in such languages.

For GPT text-davinci-003, we observe the same pattern when comparing supervised and unsupervised-LDP. It is interesting to see that GPT's scores for Indic languages are lower than BLOOM but higher for African languages, despite the fact that the African languages are likely to have less data coverage. One reason for this may be the token fragmentation issue, which we will elaborate in Section 4.4. Similarly, we observe that LDP performs competitively with supervised prompting on 20 European languages with LLaMA (Touvron et al., 2023), which we detail in Table 7 in the Appendix.

## 4.2 Non-English-centric Translation

For non-English $X \rightarrow Y$ directions, we compare our unsupervised method $\mathcal{F}^{mt}_{X \rightarrow Y}$ with supervised prompting in three categories: High-High resource languages with Vi and Fr, High-Low resource between Zh, Es, Ne (Nepali) and Pa (Punjabi), and Low-Low resource languages with Sw (Swahili), Ta (Tamil) and Te (Telugu). We use the same model and evaluation pipelines as explained in Section 4.1. For this experiment, we evaluate on the devtest sets provided by Costa-jussà et al. (2022).

---

[1]BLOOM:huggingface.co/bigscience/bloom. GPT:openai.com. We conduct full-set evaluations for 4 random languages in each group and observe < 1chrF++ standard deviation from our 200-sample evaluations.

[2]The synthetic outputs themselves are still lower-quality than supervised translations or the ground truths.

Table 3: ROUGE-L / GPT-4-EVAL scores (1-5 ratings) of different prompting techniques using InstructGPT text-davinci-003 for zero-shot summarization in high-resource (Es, Vi, Id) and low-resource (Sw, So, Mr) in the Extreme-summarization (X-sum) task (Narayan et al., 2018).

| davinci-003 | Es | Vi | Id | Sw | So | Mr |
|---|---|---|---|---|---|---|
| Basic | 12.7/2.99 | 12.6/2.77 | 12.8/2.55 | 12.2/2.33 | 11.5/3.05 | 4.1/2.98 |
| XLT | 17.7/3.90 | 14.8/3.76 | 17.6/3.40 | 20.5/3.11 | 18.5/3.96 | 10.3/3.84 |
| LDP | 18.1/4.11 | 17.4/3.76 | 18.6/3.58 | 21.8/3.32 | 19.0/3.98 | 10.0/3.89 |
| LDP+Unlabeled | 18.1/4.16 | 17.0/3.82 | 24.8/3.82 | 23.5/3.25 | 19.3/4.00 | 11.4/3.90 |

As reported in Table 2, our unsupervised LDP technique also performs on par with supervised prompting in High-High Vi-Fr pairs. More interestingly, for High-Low and Low-Low language pairs, our unsupervised method even outperforms supervised prompting for these languages by up to 5 chrF++, largely thanks to the presence of English intermediate translations in the exemplars.

## 4.3 ZERO-SHOT MULTILINGUAL SUMMARIZATION

We extend our LDP method to multilingual summarization by combining LDP with cross-lingual prompting (XLT) (Huang et al., 2023) using the instruction-tuned text-davinci-003 model. XLT is a recent English-pivoting instruction proposed by Huang et al. (2023). We follow the LDP adoptions for summarization with (LDP + Unlabeled or $\hat{\mathcal{F}}^{sum}$) and without (LDP or $\mathcal{F}^{sum}$) unlabeled data, as described in Section 3.3. We conduct evaluation on the Extreme Summarization benchmark (Narayan et al., 2018) in both high-resource (Es, Vi, Id-Indonesian) and low-resource (Sw, So-Somali, Mr-Marathi) languages. To avoid exceeding the model context length, we sample 100 documents with less than 1500 characters for each test set and obtain only 1 in-context exemplar via LDP. We evaluate the models with ROUGE-L (Lin, 2004) and GPT-4-EVAL (Liu et al., 2023). GPT-4-EVAL is a GPT-4 based metric that recently scores best in human judgement alignment. We compare our methods with XLT, and basic instruction. As shown in Table 3, our LDP methods outperform standard XLT across all languages by up to 7 ROUGE-L and exceeds basic prompting by large margins. Our methods are also consistently preferred by GPT-4-EVAL with higher ratings.

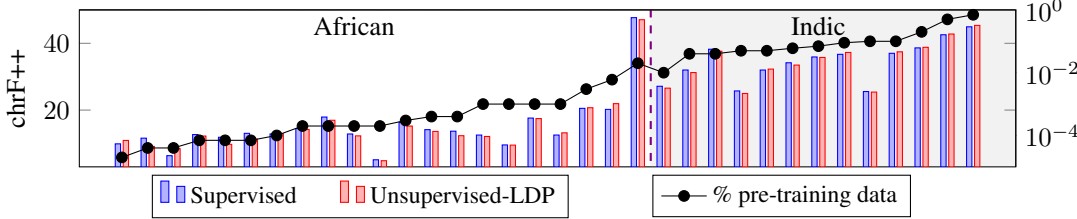

Figure 3: chrF++ scores for translation from English to each Indic and African language in the ROOTS corpus (En→X), using BLOOM. The right y-axis indicates corresponding pre-training coverage of each language at log scale.

|  | gu | mr | pa | kn | ne | te | ml | ur | ta | bn | hi | ## | chrf++ |
|---|---|---|---|---|---|---|---|---|---|---|---|---|---|
| gu | 0.7 | 0.0 | 0.0 | 0.0 | 0.0 | 0.0 | 0.0 | 0.0 | 0.0 | 0.0 | 0.0 | 0.2 | 19.64 |
| mr | 0.0 | 0.1 | 0.0 | 0.0 | 0.0 | 0.0 | 0.0 | 0.0 | 0.0 | 0.0 | 0.7 | 0.1 | 8.98 |
| pa | 0.0 | 0.0 | 0.0 | 0.0 | 0.0 | 0.0 | 0.1 | 0.0 | 0.0 | 0.0 | 0.3 | 0.6 | 1.01 |
| kn | 0.0 | 0.0 | 0.0 | 0.7 | 0.0 | 0.0 | 0.0 | 0.0 | 0.0 | 0.0 | 0.0 | 0.3 | 15.27 |
| ne | 0.0 | 0.0 | 0.0 | 0.0 | 0.6 | 0.0 | 0.0 | 0.0 | 0.0 | 0.0 | 0.4 | 0.0 | 25.28 |
| te | 0.0 | 0.0 | 0.0 | 0.1 | 0.0 | 0.8 | 0.0 | 0.0 | 0.0 | 0.0 | 0.0 | 0.1 | 19.88 |

(a) LDP without back-translation $\mathcal{F}^{mt}_{En\to X}$.

|  | gu | mr | pa | kn | ne | te | ml | ur | ta | bn | hi | ## | chrf++ |
|---|---|---|---|---|---|---|---|---|---|---|---|---|---|
| gu | 1.0 | 0.0 | 0.0 | 0.0 | 0.0 | 0.0 | 0.0 | 0.0 | 0.0 | 0.0 | 0.0 | 0.0 | 32.18 |
| mr | 0.0 | 0.9 | 0.0 | 0.0 | 0.0 | 0.0 | 0.0 | 0.0 | 0.0 | 0.0 | 0.1 | 0.0 | 18.78 |
| pa | 0.0 | 0.0 | 1.0 | 0.0 | 0.0 | 0.0 | 0.0 | 0.0 | 0.0 | 0.0 | 0.0 | 0.0 | 22.35 |
| kn | 0.0 | 0.0 | 0.0 | 1.0 | 0.0 | 0.0 | 0.0 | 0.0 | 0.0 | 0.0 | 0.0 | 0.0 | 25.82 |
| ne | 0.0 | 0.0 | 0.0 | 0.0 | 1.0 | 0.0 | 0.0 | 0.0 | 0.0 | 0.0 | 0.0 | 0.0 | 33.14 |
| te | 0.0 | 0.0 | 0.0 | 0.0 | 0.0 | 1.0 | 0.0 | 0.0 | 0.0 | 0.0 | 0.0 | 0.0 | 33.18 |

(b) LDP with back-translation $\mathcal{F}^{mtbt}_{En\to X}$.

Figure 4: Probabilities of whether the BLOOM model generates the right language for En→X task using LDP without (4a) and with (4b) intra-lingual BT prompts. Columns indicate the languages the model generates into while rows are the languages it is *supposed* to generate. ## are other languages.

Table 4: Table 4a: Impact of English tag, native language tags and no language tag for in-context prompts in Indic languages. Table 4b: Impact of different choices of LDP languages on $X{\rightarrow}$En directions using LDP without back-translation ($\mathcal{F}^{mt}$) across **10** Indic languages excluding Ta, Bn and Hi (Indic10). Note that we use supervised exemplars in table 4b for analysis purpose.

(a) Different language tags (chrF++).

| BLOOM | Indic13-En | En-Indic13 |
|---|---|---|
| **Supervised** | | |
| En-tag | 47.31 | 34.66 |
| **Unsupervised LDP** | | |
| En-tag | 46.96 | 22.53 |
| En-tag + BT | 47.43 | 34.41 |
| Native-tag | 46.90 | 29.80 |
| Native-tag + BT | 47.52 | 35.22 |
| No-tag | 46.81 | – |
| No-tag + BT | 47.62 | 34.54 |

(b) Choices of LDP languages (chrF++).

| BLOOM | Indic10-En | En-Indic10 |
|---|---|---|
| **Supervised** | 46.32 | 32.44 |
| **Unsupervised LDP with $\mathcal{Z} =$** | | |
| Ar,Zh,Vi,Fr (default) | 45.53 | 17.65 |
| Hi,Hi,Hi,Hi (Hindi) | 43.27 | 15.34 |
| Ta,Bn,Hi (Indic) | 45.51 | 16.25 |
| Fr,Es,Pt (European) | 45.31 | 18.98 |
| Vi,Vi,Vi,Vi | 44.91 | 12.94 |
| Zh,Zh,Zh,Zh | 44.71 | 15.78 |
| Ar,Fr,Es,Pt,Vi,Zh,Id | 45.50 | 16.88 |

## 4.4 ABLATION STUDY

In this section, we conduct various analysis experiments to provide a deeper understanding of our LDP method and the importance of each component.

**Breakdown of Language Pairs.** Figure 3 shows the breakdown of chrF++ performances between supervised and unsupervised-LDP promptings for each of the 34 low-resource languages. We observe that LDP performs generally on par with supervised prompting and does so equally across all languages, without unevenly performing much worse or better in any particular language. More information on performance breakdown are shown in the Appendix.

**Generating the Right Language.** Figure 4a reveals that one reason the models struggle to translate En$\rightarrow X$ when using LDP prompts $\mathcal{F}^{mt}$ (without intra-lingual BT data) is that the target-side distribution contains multiple languages, and the models struggle to recognize unfamiliar language tags, such as Marathi (Mr), and often generate wrong translations in the wrong languages (*e.g.,* Hindi instead of Marathi). Meanwhile, supplying synthetic intra-lingual prompts where the target-side is consistently in the intended language, as shown in Figure 4b with $\mathcal{F}^{mtbt}$, is more important in getting the models to recognize language rather than the language tag. In fact, we found that removing the language tag entirely can help improve the performance slightly.

**Impact of Native Language Tag.** The reason why we need unlabeled text to create intra-lingual prompts for En$\rightarrow X$ direction is because the models fail to recognize the correct language from the English language tags. A convenient way to eliminate such unlabeled text is to replace English-tag prompts (*e.g.,* "Spanish:[es-text]\nChinese:[zh-text]") with native language tags for the target language (*e.g.,* "Español:[es-text]\n中文:[zh-text]"). Such native tags serve as examples of how the intended language looks like. As shown in Table 4a, using LDP with native language tags without using any unlabeled text or intra-lingual back-translation (BT) prompts improves the performance of En$\rightarrow X$ tasks significantly, compared to using English tags. This method even approaches the performance of 8-shot supervised prompting and LDP with unlabeled BT prompts. Combining it with back-translation data (Native-tag + BT) even helps to outperform supervised prompting. In fact, the English tag may confuse the model to an extent that not using the language tag at all (*e.g.,*using "Input:[input]\nOutput:[output]") does not hurt the performances.

**Choice of LDP languages.** Another necessary question to ask is which high-resource languages should be selected as LDP exemplars. Table 4b examines which LDP language choice is optimal. As shown, for 10 Indic low-resource languages, choosing a single related language (Hindi), which is often called cross-lingual prompting (Zhang et al., 2023; Zhu et al., 2023), can be disastrous as the model tends to translate the prompt language rather than the test language. Choosing a single but distant language (Vi or Zh) yields better results, while choosing a wide variety of languages across different regions (*e.g.,* Ar,Zh,Vi,Fr) may be the optimal choice.

**Comparison with Unsupervised MT.** We also compare our method against the specialized unsupervised MT model CRISS (Tran et al., 2020) on eligible languages (Gu, Ne, Hi). As shown in Table 5, unsupervised LDP prompting with BLOOM significantly outperforms CRISS across all languages, thanks to its larger size and strong English abilities.

Table 5: Comparison in chrF++ between unsupervised LDP prompting and specialized unsupervised MT CRISS (Tran et al., 2020)

|  | Gu-En | | Ne-En | | Hi-En | |
|---|---|---|---|---|---|---|
|  | → | ← | → | ← | → | ← |
| CRISS | 41.88 | 32.41 | 37.64 | 28.17 | 51.23 | 42.29 |
| **BLOOM Prompting** | | | | | | |
| Supervised | 51.63 | 38.23 | 47.07 | 35.91 | 55.18 | 44.94 |
| LDP | 50.09 | 37.63 | 48.26 | 35.76 | 55.71 | 45.36 |

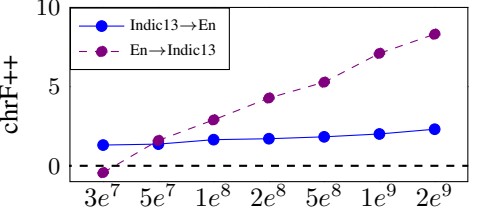

Figure 5: Gains achieved by fine-tuning BLOOM-7B w.r.t numbers of trainable parameters.

**Fine-tuning Trainable Parameters.** Figure 5 analyzes how LoRA-fine-tuned BLOOM-7B models (Hu et al., 2021) perform in $X{\to}$En and En$\to X$ Indic translation tasks as we increase the trainable parameters from 30M to 2B (full query-key-value weights). As shown, gain margins for $X{\to}$En are relatively low within 1 chrF++ as we fine-tune more parameters. Meanwhile, we observe a substantial gain of 8.7 chrF++ for En$\to X$ task, suggesting that learning to generate an unfamiliar language needs much more parameters, rendering parameter-efficient methods, like LoRA, ineffective.

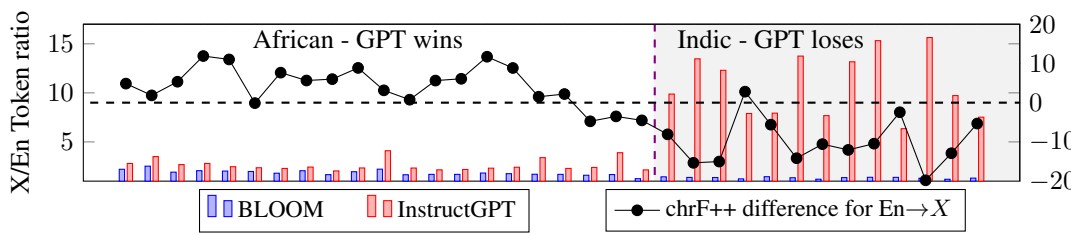

Figure 6: Fragmented tokenization issue. Left y-axis bar chart: Token length ratios between texts of language $X$ over their English counterparts of the same meaning. Right y-axis line chart: chrF++ difference between text-davinci-003 and BLOOM for En$\to X$ tasks, $> 0$ indicates instructGPT wins.

**BLOOM vs. InstructGPT.** While much evidence show that InstructGPT is superior to the vanilla BLOOM (Scao et al., 2022; Ouyang et al., 2022), our experiments in section 4.1 demonstrate it is not always true for low-resource non-Latin languages. Figure 6 explains one reason is that GPT's tokenizer fragments low-resource non-Latin texts, like Indic texts, into byte-level tokens instead of meaningful sub-word tokens. This causes an otherwise semantically short text to be stretched into a very long tokenized sequence. To take InstructGPT as an example, a 10-token English text can be equivalent to a 160-token Tamil text but only a 28-token Tumbuka text, despite Tumbuka having few data. This issue is non-existent in BLOOM, as the ratios naturally decrease when data coverage increase.

## 5 CONCLUSION

We introduce linguistically-diverse prompting (LDP), which is designed to use synthetic high-quality in-context exemplars from high-resource languages to prompt LLMs to perform generative tasks in low-resource languages. Our unsupervised approach performs on par with supervised few-shot learning for translation tasks between in English and 34 low-resource Indic and African languages, even outperforming supervised prompting in non-English-centric directions. Our method also outperforms other English-pivoting techniques in multilingual summarization.

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

## A   APPENDIX

### A.1   LOW-RESOURCE LANGUAGE DETAILS

Table 6 lists the details of each low-resource language in the ROOTS corpus (Laurençon et al., 2022) that we mainly evaluate with the BLOOM model (Scao et al., 2022). Regarding test sets, we primarily choose from the ML50 benchmark (Tang et al., 2020), which collected test data from various sources, such as WMT (Barrault et al., 2020) and FLoRes (Guzmán et al., 2019; Goyal et al., 2022). For languages absent in ML50, we choose the NLLB-devtest sets (Costa-jussà et al., 2022) as replacement. For non-English $X{\rightarrow}Y$ tasks, we choose NLLB-devtest for all our evaluation. To limit the API call costs within our budget, we randomly the same 200 samples from each test set for evaluation.

Table 6: Low-resource language details and corresponding test sets and unlabeled data sources for $X{\leftrightarrow}$En translation tasks.

| | Indic | | | | African | | |
|------|------|------|-----------|----------------|---------|----------|-----------|
| **Name** | **Code** | **Test** | **Unlabeled** | **Name** | **Code** | **Test set** | **Unlabeled** |
| Assamese | as | NLLB | CC100 | Tumbuka | –/tum | NLLB | OUR |
| Oriya | or | NLLB | ROOTS | Kikuyu | ki/kik | NLLB | OUR |
| Gujarati | gu | ML50 | CC100 | Bambara | bm/bam | NLLB | MAFAND |
| Marathi | mr | ML50 | CC100 | Akan | ak/aka | NLLB | OUR |
| Panjabi | pa | NLLB | CC100 | Tsonga | ts/tso | NLLB | MMTAfrica |
| Kannada | kn | NLLB | CC100 | Southern Sotho | st/sot | NLLB | OUR |
| Nepali | ne | ML50 | CC100 | Chewa | ny/nya | NLLB | MMTAfrica |
| Telugu | te | ML50 | CC100 | Tswana | tn/tsn | NLLB | MMTAfrica |
| Malayalam | ml | ML50 | CC100 | Lingala | ln/lin | NLLB | MMTAfrica |
| Urdu | ur | NLLB | CC100 | Northern Sotho | –/nso | NLLB | MMTAfrica |
| Tamil | ta | ML50 | CC100 | Fon | –/fon | NLLB | MAFAND |
| Bengali | bn | NLLB | CC100 | Rundi | rn/run | NLLB | OUR |
| Hindi | hi | ML50 | CC100 | Wolof | wo/wol | NLLB | CC100 |
| | | | CC100 | Luganda | lg/lug | NLLB | CC100 |
| | | | CC100 | Shona | sn/sna | NLLB | CC100 |
| | | | CC100 | Zulu | zu/zul | NLLB | CC100 |
| | | | CC100 | Igbo | ig/ibo | NLLB | CC100 |
| | | | CC100 | Xhosa | xh/xho | NLLB | CC100 |
| | | | CC100 | Kinyarwanda | rw/kin | NLLB | MMTAfrica |
| | | | CC100 | Yoruba | yo/yor | NLLB | CC100 |
| | | | CC100 | Swahili | sw/swa | NLLB | CC100 |

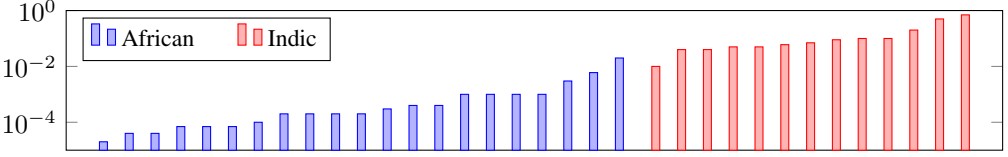

Figure 7: Low-resource language coverage % of the ROOTS corpus (Laurençon et al., 2022) used to train BLOOM. The highest-resource language for Indic and African are Hindi and Swahili. Hindi accounts for $0.7\%$ and the rarest language, Tumbuka, takes up only $2e^{-5}\%$ of the corpus.

### A.2   EXPERIMENT DETAILS

**Few-shot data sources.**   For supervised prompting, we collect randomly parallel pairs from the respective valid set for each language. For unlabeled data for our LDP method, we collect and filter data from various sources, as specified in *Unlabeled* column of Table 6. Specifically, the primary unlabeled source is the CC100 corpus (Wenzek et al., 2020; Conneau et al., 2020). For those absent in CC100, we collect data from other sources, such as the ROOTS corpus (Laurençon et al., 2022), MMTAfrica (Emezue & Dossou, 2021) and MAFAND (Adelani et al., 2022). For the remaining languages where we could not find in research repositories, we crawled from several religious and

Table 7: Comparison between supervised and unsupervised-LDP prompting with LLaMA-30B model in translation tasks between English (En) and 19 European languages (X). LDP prompts consist of exemplars from high-resource languages seen by CRISS.

| LLaMA-30B | X→En | | En→X | |
|---|---|---|---|---|
| | chrF++ | BLEU | chrF++ | BLEU |
| Supervised | 61.80 | 39.51 | 53.65 | 28.98 |
| Unsupervised-LDP | 61.75 | 38.83 | 54.00 | 29.58 |

news websites (OUR). The sizes of collected unlabeled texts vary greatly, ranging from a few millions lines for Hindi to less than 1000 lines for Bambara, thus presenting a challenge for data balancing. For LDP non-English high-resource exemplars, we randomly collect a single high-quality sentence of similar lengths from the CC100 corpus for each language and use the unsupervised CRISS model (Tran et al., 2020) to translate them into English.

**Unlabeled data filtering** To ensure high-quality native texts for unsupervised LDP prompting as well as larger-scale synthetic data creation for fine-tuning, we filter unlabeled texts such that they (*i*) are within 20 to 200 character lengths, (*ii*) do not contain non-conversational artifacts like URLs, brackets, bullet points or excessive numbers, and (*iii*) do not contain more than 20% alphabetical characters for Indic and non-latin characters for African languages. For fine-tuning, we use an upscaling temperature of 25 to smoothen the data mixture imbalance.

**Other Details.** We evaluate translation tasks with chrF++ (Popović, 2015) and SacreBLEU (Post, 2018). For SacreBLEU, we use the default tokenizer for Latin-based languages, while follow Guzmán et al. (2019); Goyal et al. (2022) to use indic_nlp_library for Indic language tokenization.

## A.3 ADDITIONAL EXPERIMENTS

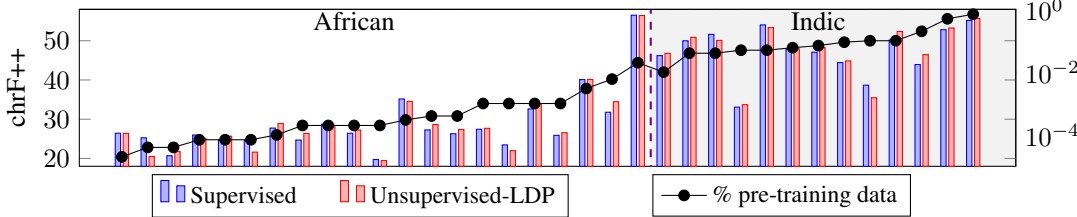

Figure 8: chrF++ scores for translation from each Indic and African language in the ROOTS corpus to English (X→En), using BLOOM. The right y-axis indicates corresponding pre-training coverage of each language at log scale.

**Breakdown of X→En.** Similar to the observation for En→X in the main paper, Figure 8 shows that LDP performs generally on par with supervised prompting equally across all languages, and that it does not unevenly perform much worse or better in any particular language.

**High-resource Translation with Llama** LLaMA (Touvron et al., 2023) is another open-sourced LLM that only supports 20 European high-resource languages. We evaluate LLaMA in translation tasks between English and the remaining 19 languages, which include Hungarian, Danish and Catalan. Specifically, we use CRISS to generate synthetic LDP exemplars from De, Es and Fr, which we then use to prompt LLaMA to translate from and to such languages. As reported in Table 7, we observe similar trends where our LDP method performs competitively with supervised prompting. The overall scores for such languages are also much higher than those of non-Latin languages because LLaMA was also pre-trained with bitexts, though without explicit alignments.

