# OpenReview forum: "Democratizing LLMs for Low-Resource Languages by Leveraging their English Dominant Abilities with Linguistically-Diverse Prompts"
_ICLR.cc/2024/Conference — Submitted to ICLR 2024_

### Official Review · Reviewer_8RdV · 2023-10-17

**Soundness:** 2 fair
**Presentation:** 3 good
**Contribution:** 2 fair
**Rating:** 5
**Confidence:** 4

**Summary:**

This paper suggests employing back-translation for few-shot, in-context learning of machine translation for low-resource languages. Initially, synthetic examples are generated by instructing the model to translate into English, utilizing few-shot, in-context learning with a variety of examples.

**Strengths:**

The idea is simple to follow and the author tested the idea across diverse set of languages. They also show improvement compared to some baselines.

**Weaknesses:**

A major weakness in this work is that the author essentially reintroduces the concept of back-translation, a well-established technique in machine translation. Yet, this paper does not make any reference to the original, popular back-translation work by Sennrich in 2016, which raises questions about the author's familiarity with prior research and the potential reinvention of existing concepts.

The primary distinction is that it is now presented in the form of a few-shot in-context learning, rather than for training purposes. One of the proposed comparisons involves fine-tuning using synthetic data in the opposite direction, which basically is the original back-translation concept. The absence of this reference is significant because it represents the core idea of this paper. See also my 2nd question.

The synthetic back-translation data was generated by providing in-context examples across a diverse set of languages (Figure 1a). However, I believe that this crucial idea is not thoroughly explored, considering the potential variability in the types of languages, examples, diversity that could be explored. Conducting an ablation study involving different examples and languages would strengthen the paper's claims. Also, while the author comments that LDP method (Figure 1c) is superior to the standard few-shot approach (Figure 1b), there is a lack of experimental results to substantiate this claim.

**Questions:**

- Is there any specific reason on choosing BLOOM over BLOOMZ (instruct-tuned version of BLOOM)? I think it will be more fair comparison vs InstructGPT.

- One of the strengths of the original back-translation approach lies in its ability to generate synthetic data at scale, and the size of the generated data can influence performance. However, I am uncertain about the data size used for your fine-tuning comparison.

---

> ### Author Response · Authors · 2023-11-11
> **Response from authors**
>
> We thank the reviewer for your thorough response.
>
> Before we clarify you concerns in the below section. We want to inform that, with new experimental results presented below, our method also works for language-understanding and pure knowledge-based question answering with XQUAD and no-context TydiQA benchmarks. We use LDP with En,Vi,Zh for XQUAD and En,Id,Ko for TydiQA.
> |XQUAD/chatgpt/F1 | Arabic | Hindi | Thai
> | --- | --- | --- | --- |
> | 0-shot | 52.91 | 45.96 | 26.34
> | 3-shot | 69.96 | 69.34 | 53.80
> | 0-shot-LDP(EnViZh) | 69.80 | 68.94 | 53.98
>
> | TydiQA/chatgpt/F1 | Arabic | Bengali | Finnish | Russian
> | --- | --- | --- | --- |  --- |
> | 0shot             | 19.17 | 5.72 | 21.79 | 12.31
> | 3-shot            | 27.78 | 20.21 | 34.72 | 16.82
> | 0-shot-LDP(EnIdKo) | 23.21 | 18.90 | 32.60 | 16.97
>
> Let us clarify your concerns:
> 1. **Our paper does not, in absolutely anyway, claim to invent back-translation!**  We cited numerous back-translation papers (Edunov et al., 2018; Lample et al., 2018; Conneau & Lample, 2019; Liu et al., 2020; Nguyen et al., 2022b). (Sennrich et al., 2016) was the first, and has been improved and refined in various works we cited above. Each of them is better than (Sennrich et al.) or complement it in other settings. **We thank the reviewer for bringing up this well-known work, and we will cite (Sennrich et al., 2016) in the paper!** But we would like to emphasize that we initially skipped its citation in the same way many papers nowadays have also stop citing (Vaswani, et al, 2017) Transformer paper due to its utmost popularity, but instead cite other improved versions, and not because we are not aware of it. We sincerely plead that missing an amendable citation is not punishable with rejection.
> 2. Our work tackles the questions of **(1) How to do back-translation with LLMs (2) How to do back-translation without any training because some LLMs are closed-source (3) How to do it with zero or just 3 unlabelled data samples from the target languages. (4) How to do it when the pretraining data for a target language is possibly less than 1MB.**
> 3. Training on synthetic back-translation data is a well-established idea and this is not our novelty claim. But how to create those data depends on settings (unsupervised or supervised), constraints (data size), tasks (summarization or translation), or target languages (high or low-resource, Latin or non-Latin), and have been explored in the many works we cited. Our work explore BT in the conditions explained above.
> 4. **Ablation study**: **Table 4a** examines the different types of prompts, language tags and with/without synthetic backtranslation. **Table 4b** explored different language choices, from the number of languages to use, which groups of languages (distant or close), and repeating the same languages. We also conduct other ablation studies in **figure 3, figure 4, table 5, figure 5, and figure 6**.
> 5. **Figure 1b vs 1c:** **Table 4a** compares Fig. 1b (without BT) vs Fig. 1c (with BT) and shows that Fig. 1c outperforms Fig. 1b greatly for En-->X direction, but observes no difference for X-->En direction. This is consistent with the explanations in figures 1a, 1b, and 1c. Please note that Fig. 1b is **not** standard supervised few-shot prompting, but our LDP method **without** synthetic intra-lingual backtranslation data.
> 6. **Why BLOOM?** Because LDP is a fully unsupervised prompt technique, as we explain in page 6, we do not want the examined models to be contaminated with supervised data and especially test sets. BLOOMz was trained on test sets! InstructGPT experiments are to show that our method works on instruction-tuned models. We also tested on GPT-3 base and observe similar results, but its performance was much worse than BLOOM in absolute terms for these low-resource languages across different settings. We explain GPTs' weaknesses in figure 6.
> 7. **Finetuning data size:** sizes are very small! **<500-1000** samples for extremely low-resource languages like Tsonga, and **~50K** samples for more popular ones like Hindi. We understand BT needs millions of samples in data size, but it was difficult to obtain such data for low-resource languages. We will put these details in paper.
>
> Given our explanations and additional experimental results, we sincerely hope the reviewer will reconsider your evaluation. If you have other questions or requests for additional experiments, we will try our best to accommodate and perform such experiments.

---

> > ### Comment · Reviewer_8RdV · 2023-11-23
> >
> > Thanks for the clarification, I bumped my score.
> >
> > Those BT papers were mainly discussed briefly in the related work, rather than being the core of the idea design and introduction. One might argue not citing Vaswani if they use some kind of more modern model as a black box (e.g., BERT), but in your case, your main idea itself is based on BT. Therefore, I would hope the authors re-clarify this in the updated manuscript.

---

> ### Author Response · Authors · 2023-11-23
> **New results on instruction-following and a humble request for re-evaluation**
>
> We thanks the reviewer for the score change and response! We deeply appreciate it. However, in our all-out effort to **win your acceptance rating**, we provide **2** more updates in our humble hope that you would consider our work if there is no critical issue.
>
> 1. We add new experiments showing that our LDP prompting method can also perform **instruction following** (like ChatGPT) with a base pre-trained model **without any supervised fine-tuning**. This contributes to the completeness of our paper (in addition to question answering, translation and summarization). Please see the details below as well as the general comment above.
>
> 2. **We follow your advice to discuss extensively about back-translation** in the camera-ready updated paper once accepted. Particularly, we will rewrite and/or add the following:
>
> **In the introduction:**
>
> Our LDP prompting technique is a simple application of back-translation (Sennrich et al., 2016) for the translation tasks under the extreme constraint where we can only use a general LLM without substantial amounts of parallel or unlabeled data for the low-resource languages, and especially where we cannot train or fine-tune the LLM if it is closed-source (OpenAI's GPTs) or too large (~175B params).
>
> **In the related work:**
>
> Our LDP prompting technique applies the concept of back-translation (Sennrich et al., 2016) in a novel situation of low-resource unsupervised translation, where a general pre-trained language model is used on low-resource languages without any further training. There are major distinctions between our method and standard back-translation (Sennrich et al., 2016). The first one is that standard back-translation requires two backward and forward supervised translation models to generate a huge amounts synthetic data from unlabeled target-language data. This renders the method inapplicable in our setup because only one generic language-model is available and no parallel or huge unlabeled data is available. The second difference is that our LDP method does not require any training to work well and use less than 8 unlabeled data samples. The third one is that our LDP method only use one-way back-translation (X->En) for both En->X and X->En translation directions, in order to avoid poor generative performance in low-resource languages.
>
> **New results for instruction-following tasks**
>
> We show that our LDP prompts can also do **instruction-following** (like ChatGPT) using the base pre-trained model **WITHOUT any supervised fine-tuning or RLHF**. Along with other tasks like **question answering, translation and summarization** that we have added in the paper and author-responses, we show that our LDP method is universally effective for low-resource languages in many kinds of tasks.
>
> For this experiment, we obtain ~200 instructions across different task-types and categories (task solving, natural questions) and translate them into Vietnamese and Indonesian and test them with Llama-2 chat and base models. We use GPT-4 to evaluate the chat responses following [MT-bench framework (Zheng, et al., 2023)](https://arxiv.org/abs/2306.05685). We select 4 random instructions from En, Zh, Fr, Ru as LDP prompts.
>
> Scores results are shown in the attached anonymized image. We also display the results in numeric terms in the table below.
>
> [https://i.postimg.cc/xT4jwXS3/fig-ldp-bench-category-en.png](https://i.postimg.cc/xT4jwXS3/fig-ldp-bench-category-en.png)
>
> | Model & Prompting | Vi/task solving | Vi/math reasoning | Vi/instruction following | Vi/natural QA |   Id/task solving | Id/math reasoning | Id/instruction following | Id/natural QA
> | --- | --- | --- | --- | --- | --- | --- | --- | --- |
> | ChatGPT(3.5) | 7.47| 7.70| 9.42| 9.05| 7.85| 7.40| 9.80| 9.45
> | Llama2-13B-Chat | 6.45| 3.50| 6.15| 4.95| 5.45| 4.25| 7.67| 5.65
> | Llama2-13B-**Base** w/ *LDP* | 3.87 | 1.94 | 4.65 | 4.80 | 2.61 | 1.75 | 7.05 | 6.10
> | Llama2-13B-Chat w/ *LDP* | 3.83 | 2.78 | 8.57 | 4.94 | 6.54 | 3.06 | 8.72 | 6.85
>
> As shown, our method can invoke relatively good instruction following capability in Vi and Id even with a **BASE** model. With Llama-2-chat, which has been trained with SFT and RLHF, our method can further improve the performance in various benchmarks.

---

### Official Review · Reviewer_y5rc · 2023-10-31

**Soundness:** 3 good
**Presentation:** 3 good
**Contribution:** 3 good
**Rating:** 6
**Confidence:** 3

**Summary:**

This paper introduces ”linguistically diverse prompting” (LDP), a method aimed at improving prompt-based generative task performance in languages for which there are no available few-shot exemplars. In this method, few-shot in-context learning is enabled through leveraging in-context exemplars from various (higher-resource) languages to ”locate the task”. The authors find that their approach achieves at least comparable performance w.r.t. supervised methods for translation and multilingual summarization.

**Strengths:**

1. The presented method is well-motivated by the observations from the literature and addresses a concrete problem in low-resource NLP (i.e. lack of in-context exemplars for some languages).

2. Evaluation is rigorous and the analyses (section 4.4) provide valuable insights for this line of research.

**Weaknesses:**

Summary of Weaknesses

1) Flawed linguistic diversity: this paper claims linguistic diversity mainly on the basis of selecting languages with different scripts: ”to ensure diversity … characters are used” (page 1). However, this is not done systematically (the authors ”include various script types” and later mention ”dissimilar lexical and regional characteristics” but do not explain the exact selection process). Moreover, this misses important aspects of linguistic diversity that are captured by for instance taking into account phylogeny.

2) This paper has reproducibility issues. The results in the paper cannot be replicated, as the approaches are evaluated on 200 randomly sampled sentences from the test set, while there is no explanation or source provided that details which sentences are included or how to reproduce this selection (e.g. which random seed). Random data selection is also used in one of the baselines, namely supervised prompting (A.2) without providing details.

**Questions:**

1. Is LDP truly an unsupervised prompting method, or are some aspects more like obtaining data for a kind of weak supervision?

2. What are your criteria for distinguishing high-resource from low-resource languages?

3. In what way does improving prompting performance for certain low-resource scenarios ’democratize’ LLMs (title)?

---

> ### Author Response · Authors · 2023-11-11
> **Response from authors**
>
> We thank the reviewer for your thorough response.
>
> Before we clarify you concerns in the below section. We want to inform that, with new experimental results presented below, our method also works for language-understanding and pure knowledge-based question answering with XQUAD and no-context TydiQA benchmarks. We use LDP with En,Vi,Zh for XQUAD and En,Id,Ko for TydiQA.
> |XQUAD/chatgpt/F1 | Arabic | Hindi | Thai
> | --- | --- | --- | --- |
> | 0-shot | 52.91 | 45.96 | 26.34
> | 3-shot | 69.96 | 69.34 | 53.80
> | 0-shot-LDP(EnViZh) | 69.80 | 68.94 | 53.98
>
> | TydiQA/chatgpt/F1 | Arabic | Bengali | Finnish | Russian
> | --- | --- | --- | --- |  --- |
> | 0shot             | 19.17 | 5.72 | 21.79 | 12.31
> | 3-shot            | 27.78 | 20.21 | 34.72 | 16.82
> | 0-shot-LDP(EnIdKo) | 23.21 | 18.90 | 32.60 | 16.97
>
> Let us clarify your questions:
> 1. **Results are reproducible!** We stated that in the paper **"For each of the 68 language pairs, we sample randomly and evaluate 200 sentences from each test set with the same seed to limit the cost of API calls"** . The seed is **0**! We also used seed **0** to collect the prompts for supervised prompting (A.2). In short, **every test set in the paper is the same across all experiments and baselines**.
> 2. **Linguistic diversity**: As explained in the paper, the name "linguistically diverse" simply means we use prompts from different languages and distant from the target language, and not going any deeper or further into sophisticated aspects of linguistics, such as phylogeny.
> 3. *Selecting the optimal languages* is a combinatorial search problem with search space expands in factorial order! The paper's main point is to use prompts from diverse languages for many downstream tasks, while leaving the optimal search problem for future work. Thus, we only conduct experiments with languages choices based on common knowledge, such as "English is different from Chinese". We will amend the paper to reflect this point.
> 4. **Table 4b explains the guidelines for language selection process empirically**. Our method work best if all languages in prompts are *different from each other* and **should be distant from the target language** we want to solve tasks in.
> 5. Other than that, **as we have tried different combinations of language choices**, there is no significant difference in terms of performances. The choice of Ar,Zh,Vi,Fr is totally a random decision by the authors at the beginning. As we tried other language sets and experiments in Table 4b, we believe will get roughly the same results as long as the guidelines above are followed.
>
> For questions:
> 1. LDP itself is **truly a fully and completely unsupervised** method, both from prompting or data collection perspectives. The only thing that makes the entire process not *fully unsupervised*  is which underlying model is used. InstructGPT is not an unsupervised model and we do not know which data it was trained on. Still, we demonstrate that our method can improve InstructGPT zero-shot performances for the target language where we do not have any supervised data at hand!
> 2. Which language is high or low resource is subjective. In our paper, we assume language with < 0.1% pretraining data or <1GB data size from the CC100 corpora as very low-resource.
> 3. **Democratizing**: our method is a simple strategy that can boost LLM's performance for low-resource languages that we would not have been able to achieve with naive zero-shot prompting, and without supervised data from those low-resource languages. For example, practitioners can build a simple summarization app for Swahili-speaking community, without ever needing to collect any supervised Swahili data, which would be hard to collect.
>
> In this way, under-represented communities can benefits from existing LLMs without needing to invest in human annotations, training computations.

---

> > ### Author Response · Authors · 2023-11-23
> > **New results: Our LDP method can also perform instruction following (chatbot) with a base pretrained model without supervised fine-tuning.**
> >
> > We provide even more experiments in instruction-following that we hope would make our paper stronger. We hope the reviewer would take a look to have a more comprehensive evaluation.
> >
> > We show that our LDP prompts can also do **instruction-following** (like ChatGPT) using the base pre-trained model **WITHOUT any supervised fine-tuning or RLHF**. Along with other tasks like **question answering, translation and summarization** that we have added in the paper and author-responses, we show that our LDP method is universally effective for low-resource languages in many kinds of tasks.
> >
> > For this experiment, we obtain ~200 instructions across different task-types and categories (task solving, natural questions) and translate them into Vietnamese and Indonesian and test them with Llama-2 chat and base models. We use GPT-4 to evaluate the chat responses following [MT-bench framework (Zheng, et al., 2023)](https://arxiv.org/abs/2306.05685). We select 4 random instructions from En, Zh, Fr, Ru as LDP prompts.
> >
> > Scores results are shown in the attached anonymized image. We also display the results in numeric terms in the table below.
> >
> > [https://i.postimg.cc/xT4jwXS3/fig-ldp-bench-category-en.png](https://i.postimg.cc/xT4jwXS3/fig-ldp-bench-category-en.png)
> >
> > | Model & Prompting | Vi/task solving | Vi/math reasoning | Vi/instruction following | Vi/natural QA |   Id/task solving | Id/math reasoning | Id/instruction following | Id/natural QA
> > | --- | --- | --- | --- | --- | --- | --- | --- | --- |
> > | ChatGPT(3.5) | 7.47| 7.70| 9.42| 9.05| 7.85| 7.40| 9.80| 9.45
> > | Llama2-13B-Chat | 6.45| 3.50| 6.15| 4.95| 5.45| 4.25| 7.67| 5.65
> > | Llama2-13B-**Base** w/ *LDP* | 3.87 | 1.94 | 4.65 | 4.80 | 2.61 | 1.75 | 7.05 | 6.10
> > | Llama2-13B-Chat w/ *LDP* | 3.83 | 2.78 | 8.57 | 4.94 | 6.54 | 3.06 | 8.72 | 6.85
> >
> > As shown, our method can invoke relatively good instruction following capability in Vi and Id even with a **BASE** model. With Llama-2-chat, which has been trained with SFT and RLHF, our method can further improve the performance in various benchmarks.

---

> > ### Comment · Reviewer_y5rc · 2023-12-04
> > **Response to author rebuttal**
> >
> > Thanks for responding to my comments/questions. I believe my scores are still appropriate.

---

### Official Review · Reviewer_qSbo · 2023-11-01

**Soundness:** 2 fair
**Presentation:** 2 fair
**Contribution:** 2 fair
**Rating:** 6
**Confidence:** 3

**Summary:**

In this paper the authors tried to improve the LLMs performance on low-resource languages by creating synthetically diverse prompts in high resource languages. The authors show the effectiveness of their approach in translation and summarisation tasks.

**Strengths:**

It is an interesting approach to get good performance on Low-Resource set up. The experiments are promising.

**Weaknesses:**

The results seems promising. The authors should provide more details about:

(a) How the diverse language sets are selected? Do they observe any correlation on linguistically similar language selection vs a random set of languages?

(b) Did they study the relation of number of languages to be selected and number of examples in the prompt?

(c) Were the prompt set fixed for every test instance?

Also, it would be interesting to see the performance difference of selecting prompts from diverse languages vs creating synthetic prompts for just the pair of languages of interest.

**Questions:**

The paper would be sound if the authors can explain / provide experimental evidences on prompt selection as pointed out in the previous section.

---

> ### Author Response · Authors · 2023-11-11
> **Response from authors**
>
> We thank the reviewer for your thorough response.
>
> Before we clarify you concerns in the below section. We want to inform that, with new experimental results presented below, our method also works for language-understanding and pure knowledge-based question answering with XQUAD and no-context TydiQA benchmarks. We use LDP with En,Vi,Zh for XQUAD and En,Id,Ko for TydiQA.
> |XQUAD/chatgpt/F1 | Arabic | Hindi | Thai
> | --- | --- | --- | --- |
> | 0-shot | 52.91 | 45.96 | 26.34
> | 3-shot | 69.96 | 69.34 | 53.80
> | 0-shot-LDP(EnViZh) | 69.80 | 68.94 | 53.98
>
> | TydiQA/chatgpt/F1 | Arabic | Bengali | Finnish | Russian
> | --- | --- | --- | --- |  --- |
> | 0shot             | 19.17 | 5.72 | 21.79 | 12.31
> | 3-shot            | 27.78 | 20.21 | 34.72 | 16.82
> | 0-shot-LDP(EnIdKo) | 23.21 | 18.90 | 32.60 | 16.97
>
> Let us clarify your questions:
> 1. As demonstrated in **table 4b**, our method work best if all languages in prompts are different from each other and **must be very distant from the target language** you want to solve. Other than that, **as we have tried different combinations of language choices**, there is no significant difference in terms of performances.
> 2. Number of languages or number of examples are limit by the context length, which is around 6-8. **table 4b** also shows that there is no difference if we put 3 (Fr,Es,Pt) or 7 (Ar,Fr,Es,Pt,Vi,Zh,Id) languages in the prompt. Meanwhile, repeating the same languages (Zh,Zh,Zh,Zh) may hurt performance.
> 3. Yes, we use the **same set of prompts for every test instance, across all languages**. For selecting back-translation data, we also use the same **seed 0** to randomly select data from the train/valid set. For test set, we also use seed 0 to select 200 samples to ensure **all 200 samples being tested are exactly same!**
> 4. *Diverse vs Synthetic intra-lingual Prompts*: **table 4a** explains the requested results, indicated as LDP *without* vs *with* intralingual back-translation prompts. In short, there is no difference between them for X-->En direction, but ones with synthetic back-translation significantly outperforms the other and match supervised prompting for En-->X direction.
> 5. Prompt selection: We randomly pick unlabeled sentences from CC25 dataset for the diverse languages, and used the same ones through the entire paper. We have to tried different random selections and found results are on average the same. As indicated in the paper that the setting is about low-resource languages, so we assume no optimal prompt selection technique is used and **random selection** is the only default choice.

---

> > ### Comment · Reviewer_qSbo · 2023-11-23
> > **Reply to authors**
> >
> > Thanks for the clarification. I have updated the score.

---

> > > ### Author Response · Authors · 2023-11-23
> > > **Update with new results on instruction-following (like ChatGPT) with a base pretrained model without supervised fine-tuning**
> > >
> > > We thanks the reviewer for the score change and response! We deeply appreciate it.
> > > We provide even more experiments in instruction-following that we hope would make our paper stronger. We hope the reviewer would take a look to have a more comprehensive evaluation.
> > >
> > > We show that our LDP prompts can also do **instruction-following** (like ChatGPT) using the base pre-trained model **WITHOUT any supervised fine-tuning or RLHF**. Along with other tasks like **question answering, translation and summarization** that we have added in the paper and author-responses, we show that our LDP method is universally effective for low-resource languages in many kinds of tasks.
> > >
> > > For this experiment, we obtain ~200 instructions across different task-types and categories (task solving, natural questions) and translate them into Vietnamese and Indonesian and test them with Llama-2 chat and base models. We use GPT-4 to evaluate the chat responses following [MT-bench framework (Zheng, et al., 2023)](https://arxiv.org/abs/2306.05685). We select 4 random instructions from En, Zh, Fr, Ru as LDP prompts.
> > >
> > > Scores results are shown in the attached anonymized image. We also display the results in numeric terms in the table below.
> > >
> > > [https://i.postimg.cc/xT4jwXS3/fig-ldp-bench-category-en.png](https://i.postimg.cc/xT4jwXS3/fig-ldp-bench-category-en.png)
> > >
> > > | Model & Prompting | Vi/task solving | Vi/math reasoning | Vi/instruction following | Vi/natural QA |   Id/task solving | Id/math reasoning | Id/instruction following | Id/natural QA
> > > | --- | --- | --- | --- | --- | --- | --- | --- | --- |
> > > | ChatGPT(3.5) | 7.47| 7.70| 9.42| 9.05| 7.85| 7.40| 9.80| 9.45
> > > | Llama2-13B-Chat | 6.45| 3.50| 6.15| 4.95| 5.45| 4.25| 7.67| 5.65
> > > | Llama2-13B-**Base** w/ *LDP* | 3.87 | 1.94 | 4.65 | 4.80 | 2.61 | 1.75 | 7.05 | 6.10
> > > | Llama2-13B-Chat w/ *LDP* | 3.83 | 2.78 | 8.57 | 4.94 | 6.54 | 3.06 | 8.72 | 6.85
> > >
> > > As shown, our method can invoke relatively good instruction following capability in Vi and Id even with a **BASE** model. With Llama-2-chat, which has been trained with SFT and RLHF, our method can further improve the performance in various benchmarks.

---

### Author Response · Authors · 2023-11-14
**New experiment results and request for an active discussion**

Dear The Reviewers and The Meta-reviewer,

We appreciate the time, effort and the valuable feedback you provided in evaluating our paper.

We would like to provide additional experimental results and explanations to the major concerns. We hope that these new results and explanations may make the paper deserve a **second look and evaluation** from the reviewers, for which we are grateful!

1. Our **zero-shot/unsupervised** method does not only work for translation and summarization, it also works well for comprehension and world-knowledge **question answering** with the XQUAD and no-context TydiQA benchmarks respectively. As shown below, our method improves zero-shot and rivals 3-shot QA tasks. We use En, Vi, Zh as LDP languages for XQUAD and En, Id, Ko for TydiQA.

| -  | XQUAD |  |  | TydiQA |  |  |  |
| --- | --- | --- | --- | --- | --- | --- | --- |
| chatgpt/F1 | Arabic | Hindi | Thai | Arabic | Bengali | Finnish | Russian
| 0-shot      | 52.91 | 45.96 | 26.34 | 19.17 | 5.72 | 21.79 | 12.31
| 3-shot      | 69.96 | 69.34 | 53.80 | 27.78 | 20.21 | 34.72 | 16.82
| 0-shot-LDP  | 69.80 | 68.94 | 53.98 | 23.21 | 18.90 | 32.60 | 16.97

Our method also works on 13 **unseen** languages by BLOOM, like cz, de, kh, ko...
| BLOOM/chrF++ | Unseen-En | En-Unseen
| -- | -- | -- |
| 4-shot     | 42.76	| 28.91
| 0-shot-LDP | 42.29	|	28.80

2. There is a concern about an optimal process to select the diverse language. This is a exponential search problem with search space expands in factorial order. Our paper's main contribution is to **show that prompts from diverse languages for many downstream tasks in low-resource languages, while providing a theoretical proof for such process may be beyond the scope and limited space of the paper**. Thus, we have provided **Table 4a** and **Table 4b** to answer this question **empirically**!

The results suggest that *default* approach is to **select languages with diverse script types across the world, so that it will be effective equally for every language in the world**. Otherwise if the languages are concentrated or related, it may benefit one type of languages, but hurt another. In Table 4b, for Indic languages, using the most related language, like Hindi, as prompt, is detrimental for languages like Assamese. Meanwhile, which changing the language choices may not have significant differences. To demonstrate this, we extend Table 4b with new results in **bold** for Assasmese and Italian.

|Table-4b, BLOOM | Indic10-En | En-Indic10 | As-En | En-As | It-En | En-It
| --- | --- | --- | --- | --- | --- | --- |
|Supervised             | 46.32 | 32.44 | **46.21**	|	**27.13** | **56.80** | **45.09**
|Unsupervised LDP (without BT)
|Ar,Zh,Vi,Fr (default)  | 45.53 | 17.65 | **46.15**	|	**15.93** | **52.61** | **32.17**
|Ta,Bn,Hi (Indic)       | 45.51 | 16.25 | **46.43**	|	**15.44** | **51.35** | **30.12**
|Fr,Es,Pt (European)    | 45.31 | 18.98 | **45.12**	|	**15.61** | **52.42** | **31.33**
|Fr,Es,Pt,Vi,Id (Latin) | **44.32** | **16.22** | **46.05**	|	**12.86** | **52.44** | **34.02**
|Vi,Zh,Id (Asian)       | **45.46** | **17.32** | **46.52**	|	**15.69** | **52.75** | **32.35**
|Ar,Fr,Es,Pt,Vi,Zh,Id (All) | 45.50 | 16.88 | **46.28**	|	**15.17** | **52.63** | **34.73**
|Hi,Hi,Hi,Hi (Hindi)    | 43.27 | 15.34 | **43.53**	|	 **7.19** | **48.12** | **22.41**
|Vi,Vi,Vi,Vi            | 44.91 | 12.94 | **45.87**	|	**10.95** | **49.31** | **28.67**
|Zh,Zh,Zh,Zh            | 44.71 | 15.78 | **45.10**	|	**10.56** | **49.74** | **25.69**
| Unsupervised LDP (with BT)
|Ar,Zh,Vi,Fr            | **46.36** | **32.20** | **46.78**	|	**26.57** | **56.48** | **45.28**

3. **Table 4a** also answers questions regarding comparison between diverse language prompts vs synthetic intra-lingual prompts.

4. There is an misconception that we reinvent back-translation (Sennrich et al., 2016), or claims "Training on synthetic data generated from opposite models" as our ideas. **We do not!** This method is well-known. But, **How to create such data** depends on settings (unsupervised or supervised), constraints (data size), tasks (summarization or translation), or target languages (high or low-resource, Latin or non-Latin), and have been explored in many works we cited (Edunov et al., 2018;Lample et al., 2018;Conneau el al., 2019;Liu et al., 2020;Nguyen et al., 2022). (Sennrich et al., 2016) itself is not applicable or comparable in our unsupervised low-resource setup. More concisely, our work tackles the questions of:
    1. How to do back-translation (BT) with LLMs?
    2. How to do BT without any training because some LLMs (e.g Chatgpt) are closed-source?
    3. How to do BT with zero or just 3 unlabelled data samples from the target languages, or pretraining data size is less than 1MB?

5. More importantly, back-translation cannot be used to perform zero-shot summarization or question answering problems, while our LDP method can. LDP is used in combination with back-translation only for the translation task.

---

### Author Response · Authors · 2023-11-23
**New results: Our LDP method can also perform instruction following (chatbot) with a base pretrained model without supervised fine-tuning**

Dear The Reviewers and ACs,

We conduct another experiment to show our LDP prompts can also do **instruction-following** (like ChatGPT) using the base pre-trained model **WITHOUT any supervised fine-tuning or RLHF**. Along with other tasks like **question answering, translation and summarization** that we have added in the paper and author-responses, we show that our LDP method is universally effective for low-resource languages in many kinds of tasks.

For this experiment, we obtain ~200 instructions across different task-types and categories (task solving, natural questions) and translate them into Vietnamese and Indonesian and test them with Llama-2 chat and base models. We use GPT-4 to evaluate the chat responses following [MT-bench framework (Zheng, et al., 2023)](https://arxiv.org/abs/2306.05685). We select 4 random instructions from En, Zh, Fr, Ru as LDP prompts.

Scores results are shown in the attached anonymized image. We also display the results in numeric terms in the table below.

[https://i.postimg.cc/xT4jwXS3/fig-ldp-bench-category-en.png](https://i.postimg.cc/xT4jwXS3/fig-ldp-bench-category-en.png)

| Model & Prompting | Vi/task solving | Vi/math reasoning | Vi/instruction following | Vi/natural QA |   Id/task solving | Id/math reasoning | Id/instruction following | Id/natural QA
| --- | --- | --- | --- | --- | --- | --- | --- | --- |
| ChatGPT(3.5) | 7.47| 7.70| 9.42| 9.05| 7.85| 7.40| 9.80| 9.45
| Llama2-13B-Chat | 6.45| 3.50| 6.15| 4.95| 5.45| 4.25| 7.67| 5.65
| Llama2-13B-**Base** w/ *LDP* | 3.87 | 1.94 | 4.65 | 4.80 | 2.61 | 1.75 | 7.05 | 6.10
| Llama2-13B-Chat w/ *LDP* | 3.83 | 2.78 | 8.57 | 4.94 | 6.54 | 3.06 | 8.72 | 6.85

As shown, our method can invoke relatively good instruction following capability in Vi and Id even with a **BASE** model. With Llama-2-chat, which has been trained with SFT and RLHF, our method can further improve the performance in various benchmarks. With more effective prompt selection, we believe our method may improve the performances further. We will add more languages and details once we have more time.

We sincerely hope that these results could make our paper more extensive and complete, and that we hope the reviewers and ACs would re-evaluate (in terms of rating) our effort given the comprehensiveness and the its potential impact for under-represented communities.

---

### Meta-Review · Area_Chair_L1o9 · 2023-12-07

**Metareview:**

This is a nice piece of work that extends the translation ability of a given LLM using back-translation techniques. The experimental results are provided on 13 Indic and 21 African low-resource languages. The experiments are sound and a great detail is provided in the follow ups during the review period. I appreciate the diversity of the language pool selected in this experiment, but at its core the paper does not present results that are surprising. (If this did not work and the authors had to do a significantly different set of things to make this work that would be quite interesting!) Given the strength of LLMs it is expected that providing in-context examples that improve on the performance of translation to/from these languages. In this regard I agree with the reviewers that the contribution of the paper isn't that strong, and decide to reject it.

**Justification For Why Not Higher Score:**

As mentioned the paper's contributions in terms of ideas are not strong enough to accept the paper.

**Justification For Why Not Lower Score:**

n/a

---

### Decision · Program_Chairs · 2024-01-16

Reject